# Biodegradable Scaffolds for Vascular Regeneration Based on Electrospun Poly(L-Lactide-*co*-Glycolide)/Poly(Isosorbide Sebacate) Fibers

**DOI:** 10.3390/ijms24021190

**Published:** 2023-01-07

**Authors:** Monika Śmiga-Matuszowicz, Jakub Włodarczyk, Małgorzata Skorupa, Dominika Czerwińska-Główka, Kaja Fołta, Małgorzata Pastusiak, Małgorzata Adamiec-Organiściok, Magdalena Skonieczna, Roman Turczyn, Michał Sobota, Katarzyna Krukiewicz

**Affiliations:** 1Department of Physical Chemistry and Technology of Polymers, Silesian University of Technology, M. Strzody 9, 44-100 Gliwice, Poland; 2Centre of Polymer and Carbon Materials, Polish Academy of Science, M. Curie-Sklodowska St. 34, 41-819 Zabrze, Poland; 3Joint Doctoral School, Silesian University of Technology, Akademicka 2A, 44-100 Gliwice, Poland; 4Biotechnology Centre, Silesian University of Technology, B. Krzywoustego 8, 44-100 Gliwice, Poland; 5Department of Systems Biology and Engineering, Faculty of Automatic Control, Electronics and Computer Science, Silesian University of Technology, Akademicka 16, 44-100 Gliwice, Poland; 6Centre for Organic and Nanohybrid Electronics, Silesian University of Technology, S. Konarskiego 22B, 44-100 Gliwice, Poland

**Keywords:** blood vessel regeneration, electrospun scaffolds, PLGA, poly(isosorbide sebacate)

## Abstract

Vascular regeneration is a complex process, additionally limited by the low regeneration potential of blood vessels. Hence, current research is focused on the design of artificial materials that combine biocompatibility with a certain rate of biodegradability and mechanical robustness. In this paper, we have introduced a scaffold material made of poly(L-lactide-*co*-glycolide)/poly(isosorbide sebacate) (PLGA/PISEB) fibers fabricated in the course of an electrospinning process, and confirmed its biocompatibility towards human umbilical vein endothelial cells (HUVEC). The resulting material was characterized by a bimodal distribution of fiber diameters, with the median of 1.25 µm and 4.75 µm. Genotyping of HUVEC cells collected after 48 h of incubations on the surface of PLGA/PISEB scaffolds showed a potentially pro-angiogenic expression profile, as well as anti-inflammatory effects of this material. Over the course of a 12-week-long hydrolytic degradation process, PLGA/PISEB fibers were found to swell and disintegrate, resulting in the formation of highly developed structures resembling seaweeds. It is expected that the change in the scaffold structure should have a positive effect on blood vessel regeneration, by allowing cells to penetrate the scaffold and grow within a 3D structure of PLGA/PISEB, as well as stabilizing newly-formed endothelium during hydrolytic expansion.

## 1. Introduction

Efficient vascular regeneration is one of the challenges of current medicine. The need to regenerate blood vessels may arise from various circumstances, including congenital defects, cardiovascular diseases, and accidents accompanied by hemorrhage. Due to the fact that blood vessels have a minimal regeneration potential, and autologous grafts are limited in availability, the use of artificial vascular scaffolds is inevitable [1]. An ideal vascular scaffold should exhibit biocompatibility, support the outgrowth of endothelial cells, be able to resist blood pressure, and vanish when its role is fulfilled [2]. Currently, synthetic polymers are considered as standard materials for vascular scaffolds in clinical applications [3]. In particular, polytetrafluoroethylene (PTFE) and poly(ethylene terephthalate) (PET), which are known to exhibit good stability and maintain inertness even during a long implantation period, have been widely used in humans since the 1970s. The major limitation of PTFE and PET as vascular scaffold materials is their inability to degrade. The long-term presence of non-degradable material brings the risk of a foreign body reaction and the necessity of a second surgical intervention, i.e., the removal of the scaffold [4]. It is also known that PTFE and PET have low compliance, poor elasticity, and may induce thrombogenic reactions, leading to the formation of blood clots that could block veins or arteries [5].

Recently, research efforts have been focused on using biodegradable materials in bioengineering [6], particularly for the design of vascular scaffolds. One of the most commonly used degradable polymers is polyglycolic acid (PGA). However, its rapid biodegradation (6–8 weeks) and low mechanical properties limit its functionality [7]. Another polyester, poly(lactic acid) (PLA), is a beneficial biomaterial due to its biocompatibility, biodegradability, and low cost, but it is known to release acidic by-products when it degrades [8,9]. To improve the performance of PGA and PLA, glycolide units can be co-polymerized with L-lactide units, resulting in the formation of poly(L-lactide-*co*-glycolide) (PLGA) [10]. In addition to its excellent biocompatibility and biodegradability, PLGA can be processed with the use of an electrospinning technique resulting in the formation of uniform fibers exhibiting enough mechanical strength to easily resist pressures typical for a blood flow [11]. Hence, electrospun PLGA scaffolds have been recently described by numerous researchers as promising materials for vascular regeneration [11,12].

A new interesting class of biodegradable polymers is a group of aliphatic polyesters based on a cellulose/starch-derived 1,4:3,6-dianhydro-D-hexitols (isosorbide, isomannide and isoidide). These polymers are obtained after reduction of the respective sugar (D-glucose, D-mannose, D-idose) and two subsequent intramolecular dehydration reactions. Among diol monomers, isosorbide (IS) is the most readily available due to its origin from sorbitol, which is relatively common in nature. Currently, this sugar-based diol is industrially produced on the ton scale with low price, and the main producer is Roquette Freres with an annual production capacity of 20 kt/year [13]. IS has a unique V-shaped molecular structure formed by two cis-fused tetrahydrofuran rings with an angle of 120° and two secondary hydroxyl groups. The reactivity of these two hydroxyl groups differs significantly; however, IS can be reacted to achieve a large variety of step-growth polymers such as polyesters, polycarbonates, polyamides, polyurethanes and others [13,14,15]. Polymeric derivatives of IS are good candidates for biomedical applications. Unsaturated, IS-based resins have been proposed as polymerizable compounds of injectable biomaterials for bone surgery, either biodegradable or stable in physiological environments [16,17]. In turn, aliphatic IS-based polyesters have been investigated as a main component of parenteral antibiotic delivery systems such as in-situ-forming implants [18,19]. Recent literature studies report the use of IS-based polyesters in various biomedical fields, including fibroblast cell culture applications [20], wound dressing [21], and cardiovascular scaffolds [22,23].

In our studies, we have decided to take advantage of the biocompatibility and biodegradability of PLGA, and use it together with an IS-based polyester, poly(isosorbide sebacate) (PISEB), to form an electrospun material suitable for vascular regeneration (Figure 1). The resulting PLGA/PISEB scaffolds were extensively investigated through microscopic, spectroscopic and calorimetric techniques. To verify whether PLGA/PISEB could be used as a biodegradable vascular scaffold, it was subjected to a 12-week-long hydrolytic degradation process, while its biocompatibility and pro-angiogenic effects were assessed with respect to human umbilical vein endothelial cells (HUVEC).

## 2. Results

### 2.1. Physicochemical Characterization of Electrospun Fibers

As a result of the electrospinning process, PLGA and PLGA/PISEB were arranged in a form of fibrous scaffolds. SEM micrograph of a PLGA scaffold (Figure 2A) revealed the formation of fibers with the diameters between 0.5 µm and 5.7 µm (average: 2.13 ± 0.69 µm, median: 2.25 µm). On the other hand, electrospinning of the blend of PLGA and PISEB (Figure 2B) resulted in the formation of two populations of fibers, thinner (with diameters between 0.25 µm and 3 µm, average: 1.35 ± 0.37 µm, median: 1.25 µm), and thicker (with the diameters between 3 µm and 7 µm, average: 4.66 ± 0.79 µm, median: 4.75 µm), as evidenced from the corresponding double-peaked histogram.

Chemical identity of electrospun fibers was confirmed with the use of IR (Figure 3A) and Raman spectroscopy (Figure 3B). In the IR spectra of both polyesters, a distinct band representing stretching of ester –C=O groups can be observed at 1748 cm^−1^ for PLGA and at 1733 cm^−1^ for PLGA/PISEB. Additionally, two strong peaks at 1182 cm^−1^ and 1086 cm^−1^ showing vibration of ester –C-O groups are also present in the IR spectra of the polymers [24]. The same bands are visible in the spectrum of electrospun PLGA/PISEB, confirming the presence of both polyester components in the blend. The incorporation of PISEB in PLGA/PISEB scaffolds is also confirmed by the presence of peaks at 2932 cm^−1^ and 2848 cm^−1^ representing –CH_2_– stretching of sebacic methylene groups in the polymer structure. In the PLGA spectrum, this band is weaker and slightly shifted towards higher wavenumbers. Similar observations were made by analyzing Raman spectra of PLGA/PISEB. For instance, a –C=O band appearing in the Raman spectrum of PLGA/PISEB is composed of signals coming from both PLGA (1772 cm^−1^) and PISEB (1748 cm^−1^). Raman spectrum of PLGA/PISEB is also characterized with a band at 878 cm^−1^, which is assigned to methylene groups in PLGA [25], and a band at 810 cm^−1^, which is assigned to methylene groups in PISEB [26]. 

Thermal properties of drop-casted PISEB and electrospun PLGA and PLGA/PISEB scaffolds were determined by a differential scanning calorimetry (DSC). As can be seen in Figure 4, in the second heating run, only glass transition temperatures (T_g_) of two polyesters used for electrospinning are observed, and they are relatively distant (the difference between them is ca. 50 °C). For PLGA/PISEB blend, two T_g_s are identified and they are located at a similar position to the one detected for PISEB and PLGA polymers (shifts ca. 2 °C downward for PISEB and ca. 2 °C forward for PLGA). According to polymer miscibility theory, if two T_g_s are observed and the values are close to that observed for pure components, the adhesion between them is poor and they are practically immiscible [27]. Taking also into account the existence of two population of fibers, it could be stated that the PLGA/PISEB blend is immiscible.

### 2.2. Degradation Studies

Susceptibility of PLGA and PLGA/PISEB to hydrolytic degradation was investigated by immersing specimen into a phosphate buffered saline (PBS) for a period of 12 weeks. After this time, SEM images (Figure 5A,B), average mass changes in time (Figure 5C), and IR spectra (Figure 5D,E) were collected and analyzed. After an initial period of 4 weeks when no changes in mass were observed, a rapid degradation process occurred in the case of PLGA. Even though the cumulative mass loss was equal to 16.5 ± 3.1%, the surface of PLGA scaffolds was not found to change significantly during the 12-week period. The network of fibers remained interconnected; however, the surface of individual fibers was found to show different defects, such as pores and cracks. Some of the fibers were evidently broken. The degradation kinetics of PLGA/PISEB were found to proceed in a more stable way, and the cumulative mass loss was only 6.1 ± 1.9%. The fibers disintegrated, forming a seaweed-like scaffold. Changes in the surface morphology of PLGA/PISEB fibers can be explained by a strong phase separation, since the polymers are immiscible, as confirmed by DSC analysis. Moreover, the glass transition temperature of PISEB (T_g_ = 1 °C), which was significantly lower than the temperature of hydrolytic degradation (37 °C), can cause its viscoelastic flow and, consequently, spreading of the fibers. Interestingly, the chemical structure of both PLGA and PLGA/PISEB remained unchanged, as evidenced by the fact that IR spectra collected before degradation and after 12 weeks of degradation were identical. To examine the change of the ratio between PLGA and PISEB in the PLGA/PISEB scaffold before degradation and after hydrolytic degradation, the deconvolution of a band representing stretching of ester –C=O groups typical for PLGA (1748 cm^−1^) and PISEB (1733 cm^−1^) was performed (Figure 5E—inset). The decrease in the ratio of peak areas of PLGA to PISEB confirmed the fact that it was PLGA that degraded first, and PISEB that remained in the fibrous scaffold.

### 2.3. Biocompatibility Assessment

Biocompatibility of PLGA, PISEB and PLGA/PISEB was verified with respect to human umbilical vein endothelial cells (HUVEC). After culturing the cells on the surface of investigated materials for 48 h, a cell viability assay was performed based on their metabolic activity. The results of MTT assay (Figure 6) confirmed biocompatibility of investigated materials, since relative cell viability of PISEB (91 ± 3%) and PLGA/PISEB (91 ± 4%) was only slightly lower than for a tissue culture plastic control (100%), and identical to the relative cell viability of PLGA (94 ± 3%), which is unanimously considered as a biomaterial [28,29].

### 2.4. Long-Term Live Microscopic Observations

The visible light imaging, together with green fluorescence signals collected from nuclei of seeded HUVEC cells using JuLI_FL™ apparatus during 48–72 h microscopic observations, allowed for the screening of investigated polymers for spontaneous cell proliferation and penetration of scaffold. HUVEC cells were cultured on control polystyrene plastic plates, additionally covered with collagen type IV. Such improvement allowed for better cell adhesion and vein-like in situ morphology formation, especially when cells were compared to the vascular endothelial growth factor (VEGF) stimulated panels cultured on the EGM-2 medium supplemented with a complete Bulletkit of growth factors. Under those conditions, cells changed morphology into that typical for tube-formation specimens during the neo-angiogenesis process [30]. The morphological changes in HUVEC cells presented on surfaces of materials, such as on the particular fibers, differ from control 2D monolayer cultures (Figure 7). The cell shape was more tubular and spindle-like, similar to the tube-formation assay [30].

### 2.5. Neo-Angiogenesis Markers Expression

HUVEC cells stimulated by the fibrillar structure of electrospun materials exhibited migration phenotype and an ability to form new tubules. To analyze the markers of neo-angiogenesis process, gene profiles of HUVEC cells were collected after 48 h of incubations on either PLGA, PISEB and PLGA/PISEB, or a tissue culture polystyrene control. The following markers of neo-angiogenesis process were investigated: interleukins (IL8), vascular endothelial growth factor (VEGF), matrix metalloproteinases (MMP2 or MMP9) and tissue inhibitors matrix metalloproteinase (TIMP1 or TIMP2) (Figure 8). Genotyping of HUVEC cells revealed increased expression of IL8 in the case of both PISEB and PLGA when compared with a control. Interestingly, the presence of PLGA/PISEB did not facilitate the expression of this interleukin. Moreover, the expression of TIMP2 was found to be lower for cells cultured on the surface of investigated polymers when compared with a control. Electrospun PLGA/PISEB scaffolds greatly stimulated the expression of both MMP2 and MMP9. On the other hand, this was not accompanied with the VEGF gene expression level, which was found to be the highest in the case of a control.

## 3. Discussion

The successful design of vascular scaffolds requires a well-balanced combination of particular material properties: biocompatibility, biodegradability, and mechanical robustness. In our studies, we have hypothesized that this set of properties should be exhibited by PLGA/PISEB scaffolds fabricated through the electrospinning process. Indeed, electrospun fibers are known to possess advanced mechanical properties, particularly stiffness and tensile strength, enabling them to withstand physiologically relevant vascular conditions, and making them one of the most promising materials to be used in biomedical engineering [31,32], particularly for vascular regeneration [33,34,35]. Even though not all polymers can be processed through an electrospinning method, making a blend with other “spinnable” polymers allows the fabrication of fibrous scaffolds. Consequently, PLGA/PISEB blend was used because the low T_g_ of PISEB causes the polymer to be in a viscoelastic state at room temperature. Hence, the structure of the nonwoven fabric is not preserved (collapse and fusion of fibers occurs). PISEB also has a fairly low molecular mass (19,700 g/mol), which is not suitable for electrospinning. Because PLGA and PISEB are not miscible, as evidenced from DSC analysis, the distribution of fiber diameters was found to be bimodal. Thinner fibers, characterized with the median diameter of 1.25 µm, are expected to be made of PLGA, while thicker fibers, characterized with the median diameter of 4.75 µm, are expected to be formed of both PLGA and PISEB.

Biocompatibility and biodegradability of the designed scaffolds are provided by the presence of PLGA and PISEB. Although the applicability of PLGA in cardiovascular research is indisputable [11,12], PISEB is a material that has just started to be considered for its biomedical potential. In this study, PISEB was synthesized from isosorbide and diethyl sebacate in the presence of Novozyme^®^ 435 as a catalyst, by azeotropic method described earlier [19,36]. Both IS and sebacic acid (SBA) are derived from renewable resources (starch and castor oil, respectively), and were chosen for PISEB synthesis from a toxicological standpoint. Hydrolytic degradation of PISEB in physiological conditions should result in releasing of IS and SBA. IS has a slightly lower LD_50_ value of 24.1 g/kg (rat, oral) than non-toxic D-glucose (25.8 g/kg, rat, oral) and is classified by the US Food and Drug Administration as “generally recognized as safe” (GRAS) material [37]. SBA is a natural metabolic intermediate in ω-oxidation of medium-chain to long-chain fatty acids, and has been proven to be safe in biomedical applications [38]. SBA can be completely broken down at the β-oxidation pathway into acetyl-CoA, which can enter into the Krebs cycle. 

Biological properties of PLGA/PISEB electrospun scaffold were investigated using HUVECs, which are frequently regarded as a model for vascular endothelium investigations, such as angiogenesis [39]. ISO 10993-5:2009 standard [40] enables the differentiation between biocompatible and cytotoxic materials based on the cell survivability rate, with threshold value of 70%. Since the relative cell viability of HUVECs was found to be around 90%, it can be concluded that PLGA/PISEB is a biocompatible material, particularly with respect to endothelial cells. Genotyping of HUVEC cells collected after 48 h of incubations on PISEB, PLGA and PLGA/PISEB showed a potentially pro-angiogenic expression profile. Genes of neo-angiogenesis were observed to be in the cross-talk and correlations, and a feedback loop for upregulated MMPs, together with downregulated TIMP inhibitors, resulted in tissues with invasive and migratory cell phenotypes [41,42]. Under those conditions, cells cultured on the surface of PLGA/PISEB changed morphology into that typical for tube-formation specimens during neo-angiogenesis process [30]. Interestingly, there was no correlation between the increased tubule formation and VEGF levels in PLGA/PISEB scaffolds, suggesting that the intrinsic composition of the material constitutes the driving force towards a sustained angiogenesis by enhancing a VEGF-independent pathway. Indeed, recent biomaterial engineering studies developed specific functionalisation strategies to induce a sustained angiogenic effect through integrin-mediated direct effect, such as in the case of acellular ECM-inspired matrices [43]. Although IL8 is a pro-angiogenic factor that has been revealed to facilitate the development of angiogenesis [44], it is also one of the major mediators of the inflammatory response [45]. Therefore, the downregulation of IL-8 expression should be considered a hallmark of anti-inflammatory effects of PLGA/PISEB, particularly important when compared with the high expression of IL8 as noted for both PLGA and PISEB.

What is unique about PLGA/PISEB is its degradation pathway, and to be more precise, structural changes that occur as the result of a hydrolytic degradation process. In PLGA, a rapid decrease in mass observed after an initial period of 4 weeks is a clear sign of an autocatalytic degradation process. In short, the carboxylic acid end groups. which are formed in the polymer matrix during the degradation process. are able to catalyze further hydrolysis reactions [46]. Therefore, the rate of degradation increases with the progression of the process, which is a typical behavior for aliphatic polyesters, including PLGA [47]. Unfortunately, the acidic environment that is created during the degradation of PLGA has negative effects on cells, as shown on the example of primary smooth muscle cells [48]. In the case of PLGA/PISEB, the autocatalytic reaction of PLGA is limited due to the presence of PISEB. Due to the low T_g_ (~1 °C) and relatively low average molecular weight (MW = 19,700 g/mol) of PISEB, during incubation in PBS solution at 37 °C, this polymer is in a high-elastic state, characterized by a certain mobility of the chains. In contrast, high molecular weight (165,000 g/mol) PLGA with a T_g_ of ~52 °C is in a glassy state, where the mobility of polymer chains is strongly limited. Consequently, PLGA/PISEB after 12 weeks in PBS are observed to form highly developed structures that may have positive effects on the stability of regenerated vessels. It is known that electrospun fibers with low diameters facilitate cell adhesion and spreading, while electrospun fibers with larger diameters facilitate cell proliferation [49]. Therefore, it could be expected that during the early time after implantation, the fibrous structure of scaffolds will allow cells to penetrate it and grow within the 3D structure of PLGA/PISEB. Then, with the progression of degradation process, the scaffold should become less porous to stabilize newly-formed endothelium. When the vessel is regenerated, PLGA/PISEB is expected to vanish as the degradation process is completed. Although clinical trials proved that the biodegradable vascular scaffolds are usually fully degraded in 3 years [50], a higher degradation rate (as predicted in the case of PLGA/PISEB) could be beneficial to avoid complications associated with a long-term presence of scaffolds [4]. Even though the assumed process seems to be attractive, further in vivo experimental studies are necessary to fully reveal the applicability of elestrospun PLGA/PISEB as scaffolds for vascular regeneration.

## 4. Materials and Methods

### 4.1. Synthesis of Poly(Isosorbide Sebacate) (PISEB) 

Isosorbide (IS, 98%, Acros, Geel, Belgium) was dried under vacuum at 40 °C just before synthesis. Diethyl sebacate (98%) was purchased from Sigma Aldrich, Merck KGaA, Darmstadt, Germany and used as received. Novozyme^®^ 435 (lipase B from *Candida antarctica* immobilized on crosslinked polyacrylate beads) was purchased from STREM Chemicals Inc, Newburyport, MA, USA. 

Poly(isosorbide sebacate) (PISEB) was synthesized as previously described [19,36]. Briefly, isosorbide (1.46 g, 0.01 mol) and diethyl sebacate (2.58 g, 0.01 mol) were heated in solvent mixture (cyclohexane:benzene = 6:1 *v*/*v*, 50 mL) together with Novozyme^®^ 435 (10 wt% of the monomers) used as polyesterification catalyst (Figure 1). 

Reaction mixture was refluxed with the aid of a Dean-Stark attachment for 7 days, under dry argon atmosphere. Ethyl alcohol as a by-product was removed from the reaction mixture by molecular sieves 4A placed in a Dean-Stark apparatus. The molecular sieves were replaced every 24 h. The formed polymer was dissolved in chloroform in order to separate from Novozyme^®^ 435. Then chloroform was partially evaporated by the use of a rotary-evaporator and the polymer was precipitated in methanol, filtered and dried under vacuum to constant mass (1 mmHg, 40 °C, 24 h). The expected structure of PISEB was confirmed by ^1^H NMR and ^13^C NMR (Appendix A). GPC: MW = 19,700 g/mol, D = 1.44.

### 4.2. Synthesis of Poly(L-Lactide-co-Glycolide) (PLGA)

Poly(L-lactide-*co*-glycolide) (PLGA) copolyester was synthesized via ring-opening polymerization (ROP) of cyclic monomers: L-lactide and glycolide (both HUIZHOU Foryou Medical Devices Co., Ltd., Huizhou, China). Zirconium (IV) acetylacetonate (Zr(acac)_4_) (Sigma Aldrich, Merck KGaA, Darmstadt, Germany) was used as a copolymerization initiator. The synthesis was carried out in bulk in a Teflon reactor under the argon at 130 °C for 24 h and for the next 72 h at 115 °C. The molar ratio of the initiator to the sum of moles of monomers was 1:3000. Detailed procedure for obtaining PLGA via ROP was described by Dobrzynski et al. [10]. The real composition of copolymer was determined by proton nuclear magnetic resonance (^1^H NMR) spectroscopy. The molar ratio of lactidyl to glycolidyl units in the obtained copolymer was 85:15 mol% and the monomer conversion was 96%. The expected structure of PLGA was confirmed by ^1^H NMR and ^13^C NMR (Appendix A). GPC: MW = 165,300 g/mol, D = 2.00.

### 4.3. Electrospinning of Polymers

In the present study, PLGA was used as a reference material. The solution of the copolyester for electrospinning was obtained in dichloromethane (DCM) (Sigma Aldrich, Merck KGaA, Darmstadt, Germany). The optimal concentration of the polymer in the solution allowing for obtaining smooth fibers, equal to 10 wt%, was determined by the optimization process. In this work, PLGA/PISEB blend with a weight ratio of components equal to 1:1 was the material being under the study. It has been investigated that 14 wt% of the blend in DCM was an optimal for obtaining fibers by electrospinning.

Biodegradable nonwovens were obtained by the electrospinning process. The procedure for both solutions was the same. A TL-Pro-BM electrospinning unit (Tong Li Tech, Bao An, Shenzhen, China) was equipped with two high-voltage power supplies. The first one generated positive electric potential of 13 kV and was attached to the spinneret, which was a steel needle of 0.9 mm in diameter. The second one of −3 kV of negative electric potential was attached to a collector, which was a steel mandrel of 27 mm in diameter. The rotation speed of the collector was 400 rpm. Distance between the spinneret and the collector was 13 cm. Polymer solutions were fed to the spinning nozzle by the infusion pump (PHD Ultra 4400, Harvard Apparatus, Cambridge, MA, USA) at the rate of 1 mL/h. The temperature and relative humidity of the air in the electrospinning chamber were 18 °C and 45%, respectively. 

### 4.4. Characterization of Polymers 

The molecular weight and the molecular weight distribution of PLGA and PISEB were determined by a gel permeation chromatography (GPC) using an Agilent Technologies chromatograph equipped with differential refractometer detector (MDS RI Detector, Agilent Technologies, Santa Clara, CA, USA) and calibrated with polystyrene standards. The measurements were carried out in methylene chloride (HPLC grade) as the solvent at 30 °C with flow rate of 0.8 mL/min. The structure of PLGA and PISEB was analyzed by ^1^H and ^13^C NMR Spectroscopy in CDCl_3_ using 600 MHz Bruker Avance II Ultrashield Plus (Bruker, Billerica, MA, USA) and 600 MHz spectrometer (Varian, Palo Alto, CA, USA), respectively.

The thermal behavior of polymers and electrospun fibers was studied by a differential scanning calorimetry (DSC) using a METTLER TOLEDO DSC 3 differential scanning calorimeter (Mettler Toledo, Columbus, OH, USA). The heating and cooling rates of the samples were 10 K/min under a nitrogen atmosphere. The samples were heated for the first run from −50 to 150 °C to erase the thermal history, then cooled to −50 °C and heated again for the second run from −50 to 150 °C. The DSC thermograms were recorded for the first and second heating runs. The glass transition temperature (T_g_) was measured at the second heat scanning.

Surface morphology of investigated samples was studied using a scanning electron microscope, SEM (Phenom ProX, Thermo Fisher Scientific, Waltham, MA, USA) operating at an accelerating voltage of 10 kV. To improve the quality of imaging, samples were sputter-coated with a 5 nm gold layer (20 min, 20 mA; Q150R Quorum Technologies, Laughton, UK). The diameter of electrospun fibers was measured with the use of ImageJ software (NIH). The results are presented as an average of 100 measurements ± standard deviation. Spectrochemical properties of investigated materials were analyzed using Raman spectroscopy (Renishaw InVia, Renishaw, Wotton-under-Edge, UK) in the spectral range of 500–2000 cm^−1^ and with 514 nm excitation laser, and FTIR spectroscopy (IR PerkinElmer Spectrum two with UATR diamond accessory, Waltham, MA, USA) in the spectral range between 750–3250 cm^−1^ and spectral resolution 2 cm^−1^.

### 4.5. Degradation Studies

Hydrolytic degradation of electrospun fibers was performed by soaking rectangular pieces of materials (approx. 8 mm × 10 mm × 60 µm, 8 mg) in 2 mL of a phosphate buffered saline (PBS: 0.01 M phosphate buffer, 0.0027 M potassium chloride, 0.127 M sodium chloride, pH 7.4), and keeping them at 37 °C for 12 weeks. The mass of the specimen was recorded every second week: specimens were removed from PBS, washed with deionized water, and dried until a constant weight. After 12 weeks, IR spectra and SEM micrographs of the materials were collected as described in Section 2.4.

### 4.6. Biological Characterization

The human umbilical vein endothelial cells (pooled donor, Lonza, Basil, Switzerland), HUVEC, were cultured in EGM^TM^-2 Endothelial Cell Growth Medium (Lonza) containing all supplements from EGM^TM^-2 SingleQuots^TM^ Supplement Pack (Lonza), in 75 cm^2^ cell culture bottles. Upon reaching confluency, the cells were removed from the culture bottle by aspiration of the culture medium, washing 1× with 3 mL of Dulbecco’s phosphate buffered saline (PAN-Biotech, Aidenbach, Germany), DPBS, with subsequent addition of 3 mL of 0.25% trypsin-EDTA solution (Sigma Aldrich). After 5–10 min of incubation, the trypsinization was ceased by the addition of 6 mL of complete culture medium. The cells were seeded in 12-well plates at 2 × 10^5^ per well in 2 mL of medium volume. After 48 h of incubation, the cytotoxicity of the investigated materials was assessed by the ability of viable cells to reduce a tetrazolium dye, MTT (3-[4,5-dimethylthiazol-2-yl]-2,5-diphenyltetrazolium bromide), to insoluble formazan, as described previously [51]. The cells from the 12-well plates were collected and immersed in 50 µL of MTT solution (0.05 mg/mL in phenol red- and FBS-free DMEM-F12; PAA). The MTT solution was removed after 2 h of incubation and the resulting violet formazan crystals were dissolved in 400 µL of acidic isopropanol (0.05 M HCl). Absorbance was measured at 570 nm with the use of a multi-well plate reader SYNERGY4 (BioTek Instruments, New York, NY, USA).

For long-term microscopic observations, the HUVEC cells were seeded direct on the tested materials or control polystyrene plate (Sarstedt, Numbrecht, Germany) in EGM-2 medium, supplemented with complete Bulletkit of growth factors (Lonza). After 48 h of incubation the cells were live stained for nucleus visualization (Sybr Green dye; Life technologies, Carlsbad, CA, USA), and the signals were collected in visible or FITC channels with magnification 40×, using JuLI_FL™ apparatus (NanoEntek, Seoul, South Korea).

Pro-angiogenic markers: IL8, VEGF, MMP and TIMP (RT-qPCR) gene expression assessment was performed for control cells and after 48 h of incubation with tested materials for HUVEC cell line. After cells were trypsinized and collected, total RNA was isolated using the phenol-chloroform method of extraction, by the Total RNA Isolation kit (A@A Biotechnology, Gdańsk, Poland). The efficiency of RNA isolation was assessed spectrophotometrically, and amplification of genes was performed using commercially-available kits (Real-Time 2xPCR Master Mix SYBR A; A@A Biotechnology) and set of primers for IL8, VEGF, MMP2, MMP9, TIMP1 and TIMP2 (Genomed, Warsaw, Poland) from references [41,42,52]. The quantitative PCR reaction, preceded by reverse transcription (NG dART RT kit, EURx, Gdańsk, Poland), was performed using a CFX96 Touch™ Real-Time PCR Detection System thermocycler (Bio-Rad, Hercules, CA, USA). The thermal profile of the reaction was as follows: (1)50 °C, 2 min,(2)95 °C, 4 min,(3)54 cycles of 95 °C, 45 s; 52.3 °C, 30 s.; fluorescence reading,(4)72 °C, 5 min,(5)melting curve from 52 °C to 92 °C (every 0.5 °C at 5 s),(6)incubation for every sample at 4 °C.

The calculation of the standardized value of the relative gene expression level in an unknown sample was performed with respect to controls, following the Livak’s formula [52]: R = 2^−ΔΔCt^
where: 

R—ratio of relative gene expression between target and reference genes; 

Ct—the quantification cycle (Ct); and

ΔΔCt—the difference between the quantification cycle (Ct) of target and reference genes.

## 5. Conclusions

In this paper, we have introduced a scaffold material made of poly(L-lactide-*co*-glycolide)/poly(isosorbide sebacate) (PLGA/PISEB) electrospun fibers as a platform for blood vessel regeneration. Even though the low T_g_ and low molecular mass of PISEB did not allow for the formation of pristine PISEB fibers, electrospinning in the presence of PLGA resulted in the development of robust PLGA/PISEB scaffolds with a bimodal distribution of fiber diameters, arising from different sizes of PLGA and PLGA/PISEB fibers. Over the course of a 12-week-long hydrolytic degradation process, PLGA/PISEB fibers were found to swell and disintegrate, resulting in the formation of highly developed structures resembling seaweeds. It is expected that the change in the structure of scaffolds should have a positive effect on blood vessel regeneration by allowing cells to penetrate the scaffold and grow within a 3D structure of PLGA/PISEB, as well as stabilizing the newly-formed endothelium during hydrolytic expansion. Genotyping of HUVEC cells collected after 48 h of incubations on the surface of PLGA/PISEB scaffolds showed a potentially pro-angiogenic expression profile, as well as anti-inflammatory effects of this material. Additionally, cells cultured on the surface of PLGA/PISEB changed morphology into one typical for tube-formation specimens during the process of neo-angiogenesis. Nevertheless, further experimental studies are necessary to fully reveal the applicability of elestrospun PLGA/PISEB as scaffolds for vascular regeneration.

## Data Availability

The data presented in this study are available on request from the corresponding author.

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
