# Peer review of "Biodegradable Scaffolds for Vascular Regeneration Based on Electrospun Poly(L-Lactide-co-Glycolide)/Poly(Isosorbide Sebacate) Fibers"

_ijms, 2023, doi:10.3390/ijms24021190_

Round 1

Reviewer 1 Report

Please see attached pdf

Author Response

The manuscript attempts to present a case about biodegradable scaffolds for vascular regeneration based on electrospun poly(L-lactide-co-glycolide)/poly (isosorbide sebacate) fibers. I would like to congratulate the authors because the paper is very well written in terms of presenting the experimental workflow and conclusions. The list of references can be slightly expanded, especially where the theoretical background of methods and materials are being presented. My points are analytically listed below:

Points for consideration:

  • Point 1: In line 310, the Celsius degrees temperature value (0C) does not look nice. Perhaps it’s the font type used or the conversion to pdf. Please check it throughout the manuscript.

We would like to thank the Reviewer for the careful and detailed review of our manuscript. According to the suggestion of the Reviewer, the Celsius degree notion (°C) has been corrected throughout the manuscript.

  • Point 2: In lines 328-336, the listed values do not look nice either. Try to include them in a table or illustration.
  • Point 3: In lines 349-353, the same issue exists.

Even though providing a list of signals is a standard way to describe NMR data, we have created an additional figure (Supplementary Information), in which NMR spectra for PLGA and PISEB are presented.

  • Point 4: In lines 446-448, please insert bullet points.

According to the suggestion of the Reviewer, we have inserted bullet points (lines 466-471).

  • General Point: You should include the following references (some from the Journal you are submitting to) in the theoretical background of methods, materials and general scaffold fabrication:
  • 10.3390/ijms232315057
  • 10.3390/ijms232315016
  • 10.3390/ijms232314621
  • 10.3390/ijms232314822
  • 10.3390/bioengineering9120742
  • 10.3844/ajeassp.2022.255.263

According to the suggestion of the Reviewer, five additional references have been added to the manuscript as [6], [7], [8], [31], [32]. One reference suggested by the Reviewer (10.3390/ijms232315057) has not been added to the manuscript, because of being not relevant to this study as it described virtual screening methods to find antituberculosis agents.

Reviewer 2 Report

Title: Biodegradable scaffolds for vascular regeneration based on electrospun poly(L-lactide-co-glycolide)/poly(isosorbide sebacate) fibers

Åšmiga-Matuszowicz et al. reported the production of fibers via electrospinning obtained from poly(L-lactide-co-glycolide)/poly(isosorbide sebacate) polymer solutions. The fibres were subsequently tested as biodegradable scaffold.

Although the authors performed some tests and presented corresponding results, there are still some issues that need to be addressed before further consideration.

General comment:

The Authors stated that the polymers are non-compatible and immiscible (line 144 and 243). Also, they described how the fibers are not the homogeneous in composition. Plus, cell growth and gene expression experiments are similar between PISEB, spun PLGA, and PLGA/PISEB or no clear trend is observed.

Said so, it is not clear which and where are the pros of using such PLGA/PISEB system? How the PLGA scaffold improves by adding PISEB?

Figure 2: The same two populations observed in PLGA/PISEB system are clearly visible also il PLGA mat. What changes is that in the case of PLGA/PISEB the thinner fibers are on top of the mat, while in PLGA case are in between the layers.

Line 106: The Authors stated an average fiber diameter of 2.13±0.08 microns for PLGA system. How is that possible? having such small SD with diameters ranging from 0.5 to approx. 6 microns. How was such value obtained? The same data is missing for PLGA/PISEB system.

Line 167-171: Does the deconvolution considers also a different chemical surrounding of carbonyls caused by the hydrolysis occurred during the degradation test? The profile of the peak changes accordingly to the extent of the hydrolysis.

Line 219-220: How the Authors explain such result?

Line 220-221: Considering the error bar it seems that there is no difference between PLGA, PISEB and PLGA/PISEB in the case of TIMP2.

Figure 5: Evaluating the degradation of casted PISEB would be helpful to better understand the results. Is it worth as a fibrous scaffold to lose the morphology against 10 wt% less weight loss (approx. 85% against 5%) after 12-week period? In which timescale is the scaffold supposed to fully degrade? Does it match with the supposed application?

Line 299-300: It is not clear the meaning of the sentence.

Line 302-304: “with the progression of degradation process, the scaffold should become less porous to stabilize newly formed endothelium”, how so? Which behavior the Authors expect?

Line 373-375: Even though the GPC is present within the characterization techniques the molecular weight is not reported nor discussed in the paper.

Does the mat were coated prior to SEM analysis?

Did the Authors change the PBS after each mass record? There was any need to refill the PBS since its just 2 mL? Does the Authors considered to use bigger quantities of mat to reduce the error while weighting the sample?

Round 2

Reviewer 2 Report

no further comment